# The Impact of Bioactive Molecules from Probiotics on Child Health: A Comprehensive Review

**DOI:** 10.3390/nu16213706

**Published:** 2024-10-30

**Authors:** Linda P. Guamán, Saskya E. Carrera-Pacheco, Johana Zúñiga-Miranda, Enrique Teran, Cesar Erazo, Carlos Barba-Ostria

**Affiliations:** 1Centro de Investigación Biomédica (CENBIO), Facultad de Ciencias de la Salud Eugenio Espejo, Universidad UTE, Quito 170527, Ecuador; saskya.carrera@ute.edu.ec (S.E.C.-P.); johana.zuniga@ute.edu.ec (J.Z.-M.); 2Colegio de Ciencias de la Salud, Universidad San Francisco de Quito USFQ, Quito 170901, Ecuador; eteran@usfq.edu.ec (E.T.); cerazo@usfq.edu.ec (C.E.); 3Instituto de Microbiología, Universidad San Francisco de Quito USFQ, Quito 170901, Ecuador

**Keywords:** probiotics, short-chain fatty acids, exopolysaccharide, vitamins, peptides, postbiotics

## Abstract

**Background**: This review investigates the impact of bioactive molecules produced by probiotics on child health, focusing on their roles in modulating gut microbiota, enhancing immune function, and supporting overall development. Key metabolites, including short-chain fatty acids (SCFAs), bacteriocins, exopolysaccharides (EPSs), vitamins, and gamma-aminobutyric acid (GABA), are highlighted for their ability to maintain gut health, regulate inflammation, and support neurodevelopment. **Objectives**: The aim of this review is to examine the mechanisms of action and clinical evidence supporting the use of probiotics and postbiotics in pediatric healthcare, with a focus on promoting optimal growth, development, and overall health in children. **Methods**: The review synthesizes findings from clinical studies that investigate the effects of probiotics and their metabolites on pediatric health. The focus is on specific probiotics and their ability to influence gut health, immune responses, and developmental outcomes. **Results**: Clinical studies demonstrate that specific probiotics and their metabolites can reduce gastrointestinal disorders, enhance immune responses, and decrease the incidence of allergies and respiratory infections in pediatric populations. Additionally, postbiotics—bioactive compounds from probiotic fermentation—offer promising benefits, such as improved gut barrier function, reduced inflammation, and enhanced nutrient absorption, while presenting fewer safety concerns compared to live probiotics. **Conclusions**: By examining the mechanisms of action and clinical evidence, this review underscores the potential of integrating probiotics and postbiotics into pediatric healthcare strategies to promote optimal growth, development, and overall health in children.

## 1. Introduction

Childhood is a critical period characterized by the rapid growth, development, and maturation of various physiological systems, including the gastrointestinal tract and the immune system. During this stage, maintaining a balanced gut microbiota is essential, as it plays a vital role in nutrient absorption, metabolism, immune development, and protection against pathogens [1,2]. Disruptions in the gut microbiome, such as those caused by antibiotics, poor nutrition, or infections, can lead to various health issues in children, including gastrointestinal disorders, allergies, and impaired immune responses [3]. Recent research has highlighted the potential of probiotics—live microorganisms that confer health benefits to the host—to modulate the gut microbiota and improve pediatric health outcomes [4,5].

A key aspect of probiotics’ benefits lies in the bioactive metabolites they produce. These metabolites, including short-chain fatty acids (SCFAs), bacteriocins, exopolysaccharides (EPSs), vitamins, and neuroactive compounds like gamma-aminobutyric acid (GABA), are crucial mediators of the beneficial effects associated with probiotics [6,7]. For instance, SCFAs, such as acetate, propionate, and butyrate, play significant roles in maintaining gut barrier integrity, modulating immune function, and providing energy to colonocytes, which are particularly important in infants and young children whose immune and digestive systems are still developing [8,9].

Probiotic metabolites also contribute to the regulation of inflammatory responses, which is crucial in preventing conditions such as necrotizing enterocolitis (NEC), a severe gastrointestinal disorder that predominantly affects preterm infants [10,11]. Metabolites like bacteriocins, antimicrobial peptides produced by probiotics, help in protecting against pathogenic bacteria by disrupting their cell membranes, thereby preventing infections such as antibiotic-associated diarrhea. In addition, exopolysaccharides (EPSs) produced by probiotic strains support biofilm formation, promoting long-term colonization of the gut and protection against pathogens [12,13,14].

Furthermore, these metabolites have a direct impact on nutrient absorption, a vital function during childhood, a period marked by increased nutritional demands for growth and development. For example, certain metabolites enhance the bioavailability of essential vitamins and minerals, which are critical for bone growth, cognitive development, and overall health [15]. Additionally, bioactive molecules like GABA, produced by certain probiotic strains, have been implicated in supporting neurodevelopment and emotional regulation through their interaction with the gut-brain axis [16,17].

Given the diverse and critical roles of these bioactive metabolites in promoting child health, there is growing interest in understanding their specific mechanisms of action and potential applications in pediatric care. This review aims to provide a comprehensive overview of the various metabolites produced by probiotics and their impact on child health. We will discuss the specific types of metabolites, such as SCFAs, bacteriocins, EPS, vitamins, and neuroactive compounds, and explore how these molecules contribute to gastrointestinal health, immune modulation, nutrient absorption, and neurodevelopment in children. By examining the evidence, this review seeks to highlight the potential of incorporating probiotic-derived metabolites into pediatric healthcare strategies to enhance the health and well-being of children.

## 2. Adaptive Role of Metabolites Produced by Probiotic Bacteria

From an evolutionary perspective, the ability of probiotic bacteria to produce metabolites enables them to adapt effectively to their host [18,19]. Bacteria that generate beneficial metabolites gain a survival advantage by outcompeting harmful microbes and fostering symbiotic relationships with their hosts. This evolutionary trait allows beneficial bacteria to thrive and proliferate within the gut environment, supporting host health and contributing to the stability of the microbial community [20,21].

For example, probiotic bacteria produce various metabolites and molecules as part of their complex survival strategies within the gut environment, ensuring their survival and competitiveness while significantly contributing to the host’s health [22]. They compete for space and nutrients by producing metabolites such as SCFAs and bacteriocins, which inhibit the growth of pathogenic bacteria and other microbes [23]. SCFAs, like acetate, propionate, and butyrate lower the gut pH, creating a less hospitable for harmful pathogens and favoring the growth of beneficial bacteria, thus maintaining their niche within the gut microbiota [24]. Additionally, bacteriocins, which are ribosomally synthesized antimicrobial peptides, target and kill competing bacterial strains by permeabilizing their cell membranes, ensuring the dominance of probiotic strains in their ecological niche [25].

Furthermore, EPSs, which are high-molecular-weight polymers secreted by probiotic bacteria, play a crucial role in biofilm formation. Biofilms are structured communities of bacteria encased in a self-produced matrix that adheres to surfaces that protect the bacteria from environmental stresses, including desiccation, antibiotics, and the host immune system [14]. This stable environment enhances bacterial survival and promotes long-term colonization in the gut [26].

In addition, the production of metabolites like vitamins and SCFAs fosters a symbiotic relationship with the host [27]. For example, vitamins such as B12 and folate, synthesized by bacteria like *Lactobacillus reuteri*, are essential for the host’s metabolic processes, including DNA synthesis and energy metabolism [28,29]. By providing these essential nutrients, probiotics contribute to the host’s nutritional status and overall health, creating a mutually beneficial relationship [30].

Moreover, SCFAs produced by probiotic bacteria serve as an energy source for colonocytes and play a role in maintaining the integrity of the gut barrier, which is crucial for preventing the translocation of pathogens and toxins from the gut into the bloodstream, thereby protecting the host from infections and inflammation [31]. Additionally, SCFAs have been shown to modulate the host immune system, promoting anti-inflammatory responses and enhancing immune tolerance [32]. A summary of the metabolites and their effects are illustrated in Figure 1.

In conclusion, probiotic bacteria produce these metabolites not out of altruism but as part of their survival strategies within the gut environment. The range of survival strategies employed by these bacteria is as diverse as the array of molecules they produce, which, while primarily serving their own persistence and competitiveness, ultimately have significant and beneficial effects on the host’s health.

## 3. Specific Probiotic Strains for Children’s Health

When it comes to enhancing children’s health, selecting specific probiotic strains plays a crucial role in providing targeted benefits. Research has highlighted the efficacy of particular probiotic strains in promoting gut health, bolstering the immune system, and alleviating various health issues in children [33,34]. The gut microbiota of children is composed of numerous microorganisms, each potentially contributing to child health and development. However, this section will focus on those microorganisms with the strongest scientific evidence supporting their positive effects in children, as well as those with the most detailed descriptions of the metabolites they produce.

### 3.1. Lactobacillus rhamnosus

*L. rhamnosus* is a facultative anaerobic bacterium known for its extensive genomic diversity and metabolic flexibility. Its genome is highly adaptable, featuring a broad array of genes encoding transporters and enzymes that facilitate the utilization of various carbohydrates, including lactose, glucose, galactose, and fucose. This adaptability enables *L. rhamnosus* to thrive in different niches within the gastrointestinal tract [35,36].

The bacterium’s ability to metabolize a wide range of carbohydrates is complemented by its production of key metabolites. In children, *L. rhamnosus* produces several important metabolites, including lactic acid, acetic acid, and short-chain fatty acids (SCFAs) such as butyrate. These metabolites are crucial for maintaining a low pH in the gut, which inhibits the growth of pathogenic bacteria and fosters a healthy gut environment [37].

Additionally, *L. rhamnosus* metabolizes fucose through unique pathways, resulting in the production of lactic acid, 1,2-propanediol, acetic acid, formic acid, and carbon dioxide. These metabolites are vital for energy production and biomass formation, which distinguish *L. rhamnosus* from other lactic acid bacteria [37].

*L. rhamnosus* also possesses several genes responsible for synthesizing antimicrobial peptides, including bacteriocins, which inhibit the growth of pathogenic bacteria [38]. Furthermore, it produces EPSs that are crucial for biofilm formation. These EPSs enhance the bacterium’s ability to adhere to intestinal surfaces and resist environmental stressors, such as bile salts and low pH [39].

The presence of genes encoding antioxidant enzymes, such as superoxide dismutase (SOD) and glutathione reductase, further contributes to its protective capabilities. These enzymes help protect both the bacterium and host cells from oxidative stress [39,40].

*L. rhamnosus* has also been shown to modulate host immune responses by interacting with dendritic cells and promoting the production of anti-inflammatory cytokines, such as IL-10. This immunomodulatory effect underscores its value in maintaining gut health [35,36].

Moreover, *L. rhamnosus* produces inosine, a metabolite with antioxidant, anti-inflammatory, anti-infective, and neuroprotective properties. The production of inosine is notably upregulated in *L. rhamnosus* compared to other *Lactobacillus* species, suggesting a unique metabolic profile that enhances its probiotic and postbiotic activities [41].

### 3.2. Bifidobacterium infantis

*B. longum* subsp. *infantis* is particularly adapted to the infant gut environment, largely due to its unique genomic configuration that enables the metabolism of human milk oligosaccharides (HMOs). The genomics of *B. infantis* reveal specialized adaptations for thriving in the infant gut, including unique metabolic pathways, genetic stability, and competitive advantages through bacteriocin production [42].

The genome of *B. infantis* contains a vast array of genes encoding glycosyl hydrolases and transport proteins that specifically target and degrade HMOs, providing a competitive advantage in the infant gut [43,44].

Additionally, *B. infantis* produces key metabolites, such as acetate and lactate, which have been shown to strengthen the intestinal barrier by enhancing tight junction integrity [45]. The bacterium’s genome also contains genes for producing various short-chain fatty acids (SCFAs), which serve as energy sources for colonocytes and play a role in maintaining gut homeostasis [46]. Although the genome of *B. infantis* does not directly contain genes for producing indole-3-lactic acid (ILA), its production of ILA results from its ability to metabolize tryptophan present in the growth medium, especially when grown on HMOs [47].

Furthermore, *B. infantis* stands out among *Bifidobacterium* species for its numerous bacteriocin gene clusters, including lanthipeptides and thiopeptides. These bacteriocins provide a competitive edge in colonizing the infant gut by inhibiting harmful microorganisms [48].

In addition to its gastrointestinal benefits, *B. infantis* produces inosine, a metabolite of *B. infantis*, has been shown to exert cardioprotective effects. It mitigates cardiac inflammation and cell death during ischemia/reperfusion injury by activating the adenosine A2A receptor, which reduces pro-inflammatory cytokines and supports ATP generation through the purine salvage pathway [49].

### 3.3. Streptococcus thermophilus

*S. thermophilus*, a widely recognized probiotic bacterium found in fermented dairy products, plays a pivotal role in influencing gut health and host metabolism through its diverse metabolic activities. One of its primary metabolites is lactate, produced via the glycolysis pathway [50]. Lactate is crucial for lowering the pH in the gut, which not only facilitates milk coagulation during fermentation but also modulates the colon epithelium. This modulation affects the expression of various transporters and proteins involved in cell cycle regulation, thereby positively impacting gut health and function [51].

In addition to lactate, *S. thermophilus* produces a range of amino acids, including leucine, isoleucine, proline, aspartic acid, and tryptophan. These amino acids are essential for the bacterium’s growth and significantly contribute to the sensory properties and quality of fermented milk products. The strain also generates various fatty acids and other metabolites, such as 2-hydroxybutyric acid, D-glycerol-D-galactose-heptanol, and hydra starch, which influence the flavor profile and overall characteristics of the fermentation process [52].

Metabolic pathways in *S. thermophilus* involve the breakdown and utilization of several amino acids, including cysteine, methionine, glutamate, glutamine, arginine, aspartate, asparagine, and alanine. These amino acids are integral to the synthesis of glutathione, a potent antioxidant that helps the bacterium combat oxidative stress and environmental challenges [53]. Moreover, *S. thermophilus* affects gut metabolism by altering tryptophan metabolism, leading to reduced levels of indole derivatives and increased production of serotonin. This metabolic shift has broader implications for mood regulation and gut–brain interactions [54].

Furthermore, *S. thermophilus* produces a range of antibiotic-like compounds and bactericidal proteins, such as bacteriocins, which are instrumental in reducing uremia and inhibiting the growth of pathogenic microbes. These antimicrobial properties enhance the probiotic benefits of *S. thermophilus*, making it a valuable contributor to gut health and a potent preventive measure against infections caused by harmful bacteria [55]. Overall, the metabolic versatility and probiotic attributes of *S. thermophilus* underscore its significance in both industrial applications and health-related functions.

### 3.4. Lactobacillus acidophilus

*Lactobacillus acidophilus* is a probiotic microorganism known for producing a range of metabolites that contribute to gut health, including conjugated linoleic acid (CLA), exopolysaccharides (EPSs), and bacteriocins (BACs). The production of these metabolites is influenced by several factors, such as initial pH, temperature, incubation time, yeast extract concentration, and the availability of free linoleic acid [56].

Bacteriocins, which are antimicrobial peptides produced by *L. acidophilus*, play a key role in inhibiting the growth of pathogenic bacteria [57]. Unlike conventional antibiotics, these bacteriocins have a relatively narrow spectrum of activity and can be degraded by proteases in the gastrointestinal tract. This may limit their direct antimicrobial efficacy but also reduces their potential to disrupt beneficial microbiota [58]. Certain bacteriocin-producing strains, such as *L. acidophilus* JCM1132, have been shown to alter gut microbiota composition in healthy mice, reducing inflammatory responses and potentially preventing metabolic diseases. This finding underscores the probiotic potential of bacteriocin-producing strains in modulating gut health [59].

Another important metabolite produced by *L. acidophilus* is valeric acid, a short-chain fatty acid (SCFA) that has been demonstrated to suppress the development of non-alcoholic fatty liver disease-associated hepatocellular carcinoma (NAFLD-HCC) by inhibiting specific cellular pathways [60].

*L. acidophilus* also produces several antigenic proteins, such as GroEL (HSP60), enolase, and transcription factors EF-Ts and EF-Tu. These proteins are recognized by serum IgG antibodies in children, particularly those with autoimmune conditions like type 1 diabetes and celiac disease, suggesting a complex interaction between the immune system and commensal bacteria [61].

The strain *L. acidophilus* 5e2 synthesizes exopolysaccharides composed of glucose, galactose, and glucosamine, which may promote gut health by fostering beneficial microbial communities. Additionally, Lactobacillus species produce biosurfactants such as surlactin, which have the capacity to reduce surface tension and inhibit pathogen adhesion, thereby maintaining a balanced microbiota and providing protection against infections [62,63].

### 3.5. Saccharomyces boulardii

*S. boulardii*, a probiotic yeast, is well known for its therapeutic benefits, particularly in gastrointestinal health. Recent research has expanded our understanding of the metabolites produced by *S. boulardii*, highlighting its potential applications in gastrointestinal disorders, cancer treatment, and as a microbial cell factory.

*Saccharomyces cerevisiae* var. *boulardii* synthesizes a diverse range of bioactive metabolites, including polyphenolic compounds such as vanillic acid, cinnamic acid, and phenyl ethyl alcohol, as well as essential nutrients like vitamin B6. These compounds contribute to its antioxidant capacity and provide a foundation for its anti-carcinogenic, antibacterial, antiviral, and general health-promoting properties [64,65]. Additionally, *S. boulardii* produces molecules like erythromycin and amphetamine, further enhancing its therapeutic profile.

One of the notable metabolites produced by *S. boulardii*, particularly by the strain *S. boulardii*-B508, is the *Saccharomyces* anti-inflammatory factor (SAIF). This factor has been shown to reduce the burden of *Mycobacterium intracellulare* in human macrophages by inducing apoptosis in infected cells and inhibiting IL-8 expression through the suppression of NF-κB activation, a key regulator of the human inflammatory response [66]. *S. boulardii* also synthesizes a phosphatase capable of dephosphorylating endotoxins such as the lipopolysaccharide (LPS) of *Escherichia coli*, thereby reducing their cytotoxic effects. Furthermore, it produces a 54-kDa serine protease that decreases intestinal permeability and inhibits the secretion of water and electrolytes, enhancing its ability to protect against bacterial toxins and exert anti-inflammatory effects in the gastrointestinal tract [67,68].

Moreover, *S. boulardii* generates high levels of acetic acid at 37 °C, a characteristic linked to unique mutations in the SDH1 and WHI2 genes, which are not found in *S. cerevisiae*. These genetic traits enable the yeast to thrive in acidic environments, providing resistance to gastric conditions and supporting its efficacy as a probiotic [69,70]. Additionally, *S. boulardii* produces a serine protease that cleaves *Clostridioides difficile* toxin A, stimulates the production of antibodies against this toxin, and modulates inflammatory responses by promoting anti-inflammatory molecules like peroxisome proliferator-activated receptor-gamma (PPAR-γ) [71].

In summary, the array of bioactive compounds produced by *S. boulardii*, including organic acids, enzymes, polyphenols, and proteases, underlines its diverse probiotic activities and therapeutic potential, making it a valuable tool in managing various health conditions.

## 4. Bioactive Metabolites Produced by Probiotics: Mechanisms of Action and Their Role in Enhancing Pediatric Health

Probiotics, particularly *Lactobacillus* and *Bifidobacterium* species, are increasingly used in pediatric healthcare for their benefits in managing gastrointestinal disorders and enhancing immune function [72,73]. Research suggests that the benefits of probiotics in children are generally species-specific rather than strain-specific [4]. Probiotics have proven effective in treating conditions like necrotizing enterocolitis (NEC), antibiotic-associated diarrhea, and *Helicobacter pylori* infections [74,75,76]. Beyond gastrointestinal health, probiotics support immune modulation, nutrient absorption, and anti-inflammatory responses, potentially reducing the frequency of infections, allergies, and inflammation-related conditions [77]. Strains like *Lactobacillus rhamnosus* GG have shown promise in alleviating allergic symptoms and promoting overall child health [78]. Understanding the bioactive metabolites and mechanisms of action of probiotics is crucial for their effective integration into pediatric care.

### 4.1. Immunomodulatory Metabolites

Probiotic bacteria, particularly strains of *Bifidobacterium* and *Lactobacillus*, produce a variety of immunomodulatory metabolites that play crucial roles in maintaining gut homeostasis and modulating the host immune system. These metabolites, which include short-chain fatty acids (SCFAs), bacteriocins, indole derivatives, and vitamins, have been shown to suppress inflammation, enhance microbial diversity, and improve intestinal barrier function by altering intestinal permeability and strengthening intercellular junctions [79]. Additionally, *Lactobacillus* and *Bifidobacterium* species are well known for their production of SCFAs and vitamins, which enhance intestinal barrier integrity and promote anti-inflammatory responses [80]. Moreover, gut microbial-derived metabolites such as polyamines, choline-derived compounds, and secondary bile acids have been identified as immunoregulatory molecules that specifically affect adaptive immune responses, particularly T helper 17 and regulatory T cells, thereby influencing health and disease outcomes [81].

These strains produce immunomodulatory metabolites that positively impact the immune system in children by enhancing innate immunity, regulating pro-inflammatory cytokine expression, and preventing tissue damage from excessive inflammatory responses [82,83]. Specifically, *Lactobacillus rhamnosus* GG has been shown to induce beneficial Th1 immunomodulatory effects, thus helping to manage conditions like cow’s milk allergy and atopic dermatitis by promoting IL-10 production [84].

Additionally, probiotics modulate the infant microbiota, induce immune mediator production, and influence cytokine production by intestinal cells, showcasing their ability to shape the immune response [82]. These effects are strain-specific and impact the immune system through various pathways, such as suppressing inflammation via the NF-κB pathway and enhancing phagocytic activity [85]. Furthermore, probiotics have been associated with reducing allergic reactions by downregulating Th2-related responses, inhibiting pro-inflammatory cytokine production, and modulating immune system components, ultimately promoting anti-inflammatory and immunomodulatory effects [86]. For example, lactic acid bacteria (LAB) are particularly effective in preventing allergic diseases like atopic eczema in infants by enhancing the body’s capacity to produce immune-enhancing cytokines such as interferon-gamma (IFN-γ) [87]. Specific probiotic strains also significantly increase the production of intestinal immunoglobulin A (IgA), which enhances mucosal immunity and protects against gastrointestinal infections [88]. Moreover, probiotics can influence systemic immunity by promoting the activity of natural killer (NK) cells and the differentiation of T-helper cells, both crucial for fighting infections and maintaining immune balance [89].

*Lactobacillus gasseri* TCI515, a probiotic strain, exemplifies this role by enhancing the expression of innate immunity-regulating genes and inhibiting pro-inflammatory cytokine gene expression, thereby improving innate immunity and preventing tissue damage from excessive inflammatory responses [83] Similarly, LAB fermentation of herbal medicines generates metabolites like exopolysaccharides, SCFAs, and bacteriocins, which have immunomodulatory properties and interact with the immune system, potentially boosting the innate immune response in children [90].

Overall, probiotic bacteria play a crucial role in producing metabolites that interact with and modulate the immune system, offering significant benefits for children’s health and well-being.

### 4.2. Anti-Inflammatory Metabolites

Probiotics have been studied for their potential to produce anti-inflammatory metabolites, such as short-chain fatty acids (SCFAs) like butyrate and acetate, which may modulate immune responses and reduce inflammation in children [91]. However, the evidence is mixed and often varies depending on the specific strains used, the health status of the children, and the biomarkers measured. For example, a recent study involving a combination of *Lactobacillus acidophilus* and *Bifidobacterium lactis* demonstrated a significant reduction in inflammatory markers, such as MPIF-1 and MIP-3α, in children, suggesting an anti-inflammatory effect through immune modulation [92]. Similarly, another study found that synbiotic supplementation in overweight children led to decreased levels of inflammatory markers like tumor necrosis factor-α and interleukin-6, although these effects were associated with weight reduction rather than a direct anti-inflammatory action [93,94].

In children with various diseases, probiotics have shown potential in reducing systemic inflammation, particularly in conditions such as allergies, autoimmune diseases, and severe illnesses [94]. However, a study investigating the long-term effects of *Lactobacillus paracasei* supplementation during weaning found no significant impact on metabolic and inflammatory profiles at school age, suggesting that early probiotic intervention may not have lasting anti-inflammatory benefits [95]. Thus, the effectiveness of these biotics appears to be influenced by the health status of the children, with greater benefits observed in those with specific conditions, indicating that they may be more effective for certain groups.

Given that the molecular mechanisms underlying the anti-inflammatory effects of specific probiotic molecules are not well understood, a recent in vitro study demonstrated that secreted metabolites from *Bifidobacterium infantis* and *Lactobacillus acidophilus* exert anti-inflammatory effects in immature human enterocytes by modulating genes involved in immune response, cell survival, and NF-κB signaling. These metabolites reduce IL-6 and IL-8 production, suggesting their potential to mitigate inflammation in conditions such as necrotizing enterocolitis in children [96].

Additionally, another in vitro study evaluating the anti-inflammatory effects of biomolecules in probiotics commonly used in children showed that intestinal bacterial metabolites produced by *Bifidobacterium animalis* subsp. *lactis* LKM512 can suppress TNF-alpha production in J774.1 cells stimulated by lipopolysaccharide (LPS), suggesting that consumption of yogurt containing this strain may help reduce inflammatory cytokines produced by macrophages [97].

In conclusion, while probiotics and their metabolites show promising potential as anti-inflammatory agents in children, further investigation is needed to fully elucidate the molecular mechanisms underlying these effects and to determine optimal conditions for their application. Such insights could guide the development of targeted probiotic therapies, particularly for conditions like necrotizing enterocolitis, allergies, and autoimmune diseases, ultimately improving therapeutic outcomes for pediatric populations. A summary of the role of probiotic metabolites in children’s health is shown in Figure 2.

### 4.3. Nutrient Absorption-Enhancing Metabolites

Probiotics have been shown to enhance nutrient absorption in children by producing beneficial metabolites that improve gut health and nutrient uptake. These metabolites include vitamins, minerals, and short-chain fatty acids (SCFAs), which play crucial roles in maintaining a healthy gut microbiome and improving overall health outcomes in children [4].

A recent study demonstrated that probiotics significantly increased the blood levels of vitamins and minerals such as vitamin D, vitamin A, calcium, zinc, and iron in children over a 10-week period. This finding suggests that probiotics can enhance the absorption of these essential nutrients, potentially improving the nutritional status and immunity of children [98]. Probiotics have also been shown to regulate lipid metabolism, which is crucial for nutrient absorption. In overweight or obese children, probiotics helped reduce levels of LDL cholesterol and leptin while increasing HDL cholesterol and adiponectin, indicating improved lipid profiles and metabolic health [99,100].

The administration of prebiotic-enhanced lipid-based nutrient supplements (LNSs) in undernourished infants led to a significant increase in the production of SCFAs such as acetate, butyrate, and propionate. These SCFAs are known to enhance gut health and nutrient absorption by promoting a beneficial gut microbiota composition. Similarly, the use of multi-strain probiotics in obese children increased the abundance of beneficial bacteria like *Lactobacillus* spp. and *Bifidobacterium animalis*, which are associated with improved SCFA production and lipid metabolism [101].

A recent study assessed the effectiveness of probiotics in improving health outcomes for children with severe acute malnutrition (SAM) in the Democratic Republic of Congo. The findings suggested that probiotics contributed to better health metrics, including weight gain, shorter recovery times, and overall nutritional improvement. The study concluded that incorporating probiotics into rehabilitation protocols could enhance the recovery of malnourished children [102].

In vitro approaches have shown that a mixture of *Saccharomyces boulardii*, *Lactobacillus acidophilus*, *Lactobacillus rhamnosus*, and *Bifidobacterium breve*, in combination with the enzyme amylase, disrupts pathogenic gastrointestinal biofilms, improving nutrient absorption by enhancing permeability and increasing the penetration of proteins and vitamins through intestinal cell monolayers [103].

While the evidence supports the role of probiotics in enhancing nutrient absorption through the production of beneficial metabolites, the effects can vary based on the probiotic strains used, the duration of supplementation, and the health status of the children. Further studies are needed to optimize probiotic interventions for different pediatric populations and to fully understand the mechanisms behind these benefits.

### 4.4. Gut Microbiota-Balancing and Barrier-Enhancing Metabolites

A healthy microbiota involves maintaining an appropriate diversity and abundance of beneficial microorganisms that outnumber and outcompete potentially pathogenic or harmful ones [104]. When this delicate equilibrium is altered, the microbiota becomes imbalanced, a condition known as dysbiosis [105]. In children, dysbiosis is associated with alterations of gut function such as diarrhea [106], infant colic [107] or even autoimmune and atopic diseases such as asthma or rhinitis [108].

Balancing the gut microbiota is essential for optimal digestive health in children, and probiotic metabolites contribute significantly to this balance [109,110]. The bioactive molecules produced by probiotics help maintain a healthy gut environment by improving the balance of the gut microbiota [4]. By supporting a diverse and beneficial microbial community in the gut, these metabolites promote digestive health, prevent conditions such as diarrhea and colic, and contribute to overall well-being in children.

Short-chain fatty acids (SCFAs) are primary metabolites produced during the fermentation of dietary fibers by probiotic bacteria in the colon. These SCFAs, particularly butyrate, play a critical role in modulating the composition and activity of the gut microbiota [111]. Butyrate, for instance, is known for its ability to promote the growth of beneficial bacteria such as *Faecalibacterium prausnitzii*, a bacterium associated with anti-inflammatory properties [112]. At the same time, SCFAs help lower the pH of the gut environment, creating conditions unfavorable for pathogenic bacteria [113], thereby reducing the risk of gastrointestinal infections like diarrhea and colic in children.

In addition to promoting a healthy microbial balance, SCFAs have been found to enhance the production of mucin, a key component of the gut mucus layer that provides a protective barrier against pathogens [114]. By increasing mucin production, SCFAs help to fortify the gut lining, reducing the likelihood of pathogens adhering to and invading the epithelial cells of the intestine. Furthermore, butyrate serves as a primary energy source for colonocytes [115,116], the cells lining the colon, thereby maintaining the integrity and function of the intestinal barrier [117].

Tight junctions are critical components that regulate the permeability of the gut barrier. Probiotic-derived metabolites like butyrate have been demonstrated to upregulate the expression of tight junction proteins, including claudin and occluding [118], which are essential for maintaining the integrity of the epithelial barrier. By enhancing these tight junctions, probiotics help prevent “leaky gut,” a condition characterized by increased intestinal permeability that allows toxins, microbes, and other harmful substances to enter the bloodstream, potentially triggering systemic inflammation and immune responses.

Additionally, certain metabolites produced by probiotics, such as lactate and hydrogen peroxide, have antimicrobial properties that contribute to maintaining a healthy gut barrier. Lactate, for example, can inhibit the growth of pathogenic bacteria by lowering the pH of the gut environment [119]. Meanwhile, hydrogen peroxide, produced by various microbial strains, has direct bactericidal effects against pathogens further protecting the integrity of the gut barrier [120].

## 5. Types and Mechanisms of Action of Probiotic Metabolites in Children

Probiotic metabolites, the bioactive compounds produced by beneficial microorganisms, play a crucial role in promoting and maintaining health in children. These metabolites, which include short-chain fatty acids (SCFAs), bacteriocins, exopolysaccharides (EPSs), and vitamins, among others, exert a variety of beneficial effects on the host. Understanding the types and mechanisms of action of these probiotic metabolites is essential for leveraging their full therapeutic potential. This section delves into the various types of probiotic metabolites and explores the specific mechanisms through which they exert their health-promoting effects in children. A summary of the main metabolites produced by probiotic bacteria in children is shown in Figure 3.

### 5.1. Vitamins

Probiotic bacteria play a vital role in synthesizing essential vitamins that significantly impact the overall health and development of children. These beneficial bacteria, particularly certain strains of lactic acid bacteria (LAB) and bifidobacteria, can produce various vitamins necessary for multiple physiological processes, such as growth, development, and immune function [121]. In this section, we explore the types of vitamins produced by probiotic bacteria, their impact on children’s health, and the mechanisms through which these vitamins exert their beneficial effects.

#### 5.1.1. Vitamin B Complex

Probiotic bacteria are known to synthesize several B vitamins, including B1 (thiamine), B2 (riboflavin), B3 (niacin), B5 (pantothenic acid), B6 (pyridoxine), B7 (biotin), B9 (folate), and B12 (cobalamin). These vitamins are critical for energy metabolism, DNA synthesis, red blood cell formation, and the proper functioning of the nervous system. Ensuring adequate levels of these vitamins is crucial for children’s growth and development, as they support numerous metabolic pathways and physiological processes [15,121].

Research has shown that specific probiotic strains, such as those within the genera *Lactobacillus* and *Bifidobacterium*, are highly effective at producing B vitamins. For example, *Lactobacillus fermentum* has been identified as a robust producer of folate and vitamin B12, achieving production levels of up to 801.79 μg/mL for folate [122]. These strains can enhance the nutritional profile of various foods, such as fermented dairy products, making them an excellent source of B vitamins.

Oligosaccharides, which are prebiotic fibers that serve as a food source for probiotic bacteria, have been found to enhance the vitamin-producing capabilities of certain probiotic strains. Studies indicate that the presence of oligosaccharides can increase the synthesis of B vitamins by enhancing the bacteria’s properties, such as hydrophobicity, auto-aggregation, and biofilm formation [123].

Several probiotic strains, including *Lactobacillus reuteri*, *Lactobacillus acidophilus*, *Streptococcus thermophilus*, and *Lactobacillus rhamnosus* GG, are recognized for their ability to produce essential B-group vitamins, such as vitamin B12, riboflavin, folate, and thiamine, which are vital for numerous metabolic and physiological processes [124]. For example, *L. reuteri* strains produce corrinoids related to vitamin B12, while *L. acidophilus* strains increase riboflavin levels during fermentation, enhancing the nutritional value of foods like soymilk [125,126]. Additionally, *S. thermophilus* can boost the production of bioactive folate forms [127].

Leveraging these probiotic strains’ vitamin-producing capabilities could help develop fortified, vitamin-rich functional foods, which are especially beneficial in promoting children’s growth and development.

#### 5.1.2. Vitamin K

Vitamin K, particularly in its K2 form (menaquinone), is another vital nutrient synthesized by probiotic bacteria [128]. This vitamin plays a crucial role in blood clotting, bone health, and cardiovascular function. The synthesis of vitamin K2 by gut bacteria is especially significant in children, who may have limited dietary intake of this nutrient due to selective eating habits or inadequate nutrition [129,130].

The probiotic bacterium *Bacillus clausii* has been demonstrated to produce vitamin K2, effectively correcting coagulation disorders in infants following antibiotic treatment by normalizing prothrombin levels [131]. Similarly, *Lactococcus lactis* is a known producer of vitamin K2, particularly in the context of fermented foods, where different strains produce varying amounts of the vitamin. Under specific cultivation conditions, these bacteria can enhance the delivery of vitamin K2 through extracellular vesicles [132].

Additionally, certain neonatal gut bacteria, such as *Enterobacter agglomerans*, *Serratia marcescens*, and *Enterococcus faecium*, have been identified as producers of menaquinones, a form of vitamin K2. This suggests that even in early life, the gut microbiota can contribute significantly to vitamin K production, which is essential for proper blood clotting processes [133]. However, antibiotic treatments in infants can disrupt the gut microbiota, leading to decreased vitamin K production. Studies have shown that administering probiotics like *B. clausii* can help restore normal prothrombin levels, highlighting the role of gut bacteria in vitamin K synthesis and coagulation [131].

There are far fewer bacteria known to produce vitamin K than those that synthesize B vitamins; however, as the gut microbiota of infants and children continues to be decoded, more microorganisms capable of synthesizing vitamin K are likely to be discovered, given its importance in pediatric health.

It is important to highlight that probiotics not only produce vitamins but also enhance their absorption and bioavailability. For instance, a study involving children aged 8–13 showed that probiotic supplementation significantly increased blood levels of vitamins D and A compared to a placebo group over a 10-week period, suggesting that probiotics may improve the absorption and serum concentrations of essential vitamins [98]. Another study reported improvements in gut health markers, such as increased bifidobacteria and reduced inflammatory markers, in children receiving prebiotics and vitamin supplements, demonstrating the potential of probiotics to enhance both gut health and vitamin status [134].

The ability of probiotics to synthesize and enhance the absorption of vitamins has broad implications for child health. By ensuring the production of essential nutrients, such as B vitamins and vitamin K, probiotics support various physiological processes, including energy metabolism, immune response, and bone mineralization. Furthermore, vitamins produced by commensal bacteria may influence immune responses, suggesting roles beyond basic nutrition, such as regulating gene expression and enhancing nutrient absorption [15,31]. A summary of the probiotic bacteria involved in the synthesis of vitamins is described in Table 1.

### 5.2. Short-Chain Fatty Acids (SCFAs)

Short-chain fatty acids (SCFAs) such as acetate, propionate, and butyrate are key metabolites that play a vital role in supporting gut health in children. SCFAs are essential for maintaining a healthy digestive system [136]. One of their primary functions is to nourish the cells lining the intestines, known as colonocytes [115], which are crucial for forming a robust gut barrier that prevents harmful substances from entering the bloodstream [115]. Additionally, SCFAs are critical for the development, regulation, and maturation of the immune system [137,138]. Notably, butyrate, a specific SCFA, has demonstrated anti-inflammatory properties [136], helping to protect against conditions such as colic, gastrointestinal infections, and inflammatory bowel disease [139].

SCFAs also contribute to maintaining an optimal gut environment by lowering the pH [140,141], which supports the growth of beneficial bacteria while inhibiting gut pathogens [142]. Furthermore, these molecules enhance the gut barrier by promoting the production of mucins, substances that protect the gut lining and prevent the adherence and invasion of pathogens [143].

Certain probiotic strains, such as *Lactobacillus* and *Bifidobacterium*, are known to produce SCFAs in the gut [31,144,145,146]. These beneficial bacteria are commonly found in fermented foods like yogurt or in dietary supplements specifically formulated for children. When children consume foods rich in probiotics, these bacteria proliferate in the gut and produce SCFAs by fermenting dietary fibers present in fruits, vegetables, and whole grains [147,148].

### 5.3. Antimicrobial Peptides

Antimicrobial peptides (AMPs) are small molecules that serve as natural antibiotics produced by the body and certain probiotic bacteria [149,150]. AMPs represent a promising avenue for enhancing child health by modulating the gut microbiota and providing a natural defense against pathogens [150]. These bioactive molecules, which include bacteriocins and other peptide-based antimicrobials, have been shown to selectively inhibit harmful bacteria while promoting the growth of beneficial microbes [151]. Furthermore, unlike conventional antibiotics, AMPs have a lower propensity for developing resistance, making them a safer alternative for managing pediatric infections and promoting overall gut health [152,153,154].

Although there is currently limited direct evidence for the use of probiotic-derived antimicrobial peptides (AMPs) specifically in pediatric health, several lines of research suggest their potential as novel therapeutic agents for children. Extensive studies have demonstrated the broader benefits of probiotic-derived AMPs for human health, particularly their antimicrobial, immunomodulatory, and microbiota-regulating properties [151,155,156,157]. Additionally, AMPs from other sources have proven effective in managing pediatric infections, and clinical trials have highlighted cationic antimicrobial peptides as promising alternatives for treating infections in neonates and children, especially in cases involving antibiotic-resistant pathogens [158]. Further research into the use of probiotic-derived AMPs could open up new pathways for developing innovative, safe, and effective treatments to manage infections and enhance health outcomes in children.

### 5.4. Enzymes

The enzymes produced by probiotics hold significant potential in improving children’s health by supporting essential digestive functions and enhancing nutrient absorption [159]. These beneficial microorganisms produce enzymes that break down complex nutrients, making them more accessible for a child’s developing digestive system. As children’s gut microbiota continues to evolve, the enzymatic activity provided by probiotics plays a pivotal role in promoting a balanced gut environment, which is crucial for immune function, growth, and overall well-being [160].

One of these enzymes derived from probiotics is β-glucosidase, an enzyme important for breaking down complex carbohydrates into simpler sugars, which can enhance the probiotic’s effectiveness in the gut [161]. *Bifidobacterium* species, which are early colonizers of the infant gut, produce β-galactosidase to metabolize milk-based diets. This enzyme cleaves the glycosidic bond in lactose through hydrolysis, producing the monosaccharides glucose and galactose, which are essential for energy production and growth in infants [162]. In addition to hydrolysis, β-galactosidase can transfer galactosyl units to other sugar molecules through transgalactosylation. This activity forms galactooligosaccharides (GOSs), which are beneficial prebiotics that promote the growth of healthy gut microbiota [162,163,164]. In infants with nutritional disorders such as celiac disease and cystic fibrosis, β-galactosidase activity can be affected. For example, lactase activity is significantly reduced in celiac disease, while hetero-β-galactosidase activity remains relatively stable [165].

Similarly, lacto-N-biosidase (LNBase) plays a crucial role in the digestion of human milk oligosaccharides (HMOs). This enzyme, primarily found in *Bifidobacterium* spp., facilitates the breakdown of complex sugars into simpler forms that can be utilized by the infant’s gut microbiota [160]. For instance, LNBase from *Bifidobacterium bifidum* (LnbB) is essential for the degradation of HMOs, specifically lacto-N-tetraose, into lacto-N-biose I and lactose, which is vital for the early life microbiota in infants [166].

LNBase operates via a substrate-assisted catalytic mechanism, with a unique metabolic pathway specific to lacto-N-biose I, a major core structure in HMOs [167]. Its activity is modulated by specific amino acids, such as His263, which plays a critical role in the catalytic process by altering the pKa of the acid/base residue [166]. The stability of LNBase during digestion and its ability to modulate gut microbiota composition, increasing the abundance of beneficial bacteria like *B. bifidum*, underscores its potential in alleviating infant food allergies and promoting overall gut health [160].

The enzymatic activities of probiotics contribute significantly to infant and child health by enhancing digestion, nutrient absorption, and gut microbiota composition. These findings highlight the importance of continued research into probiotic-derived enzymes as valuable tools for improving pediatric health outcomes.

### 5.5. Exopolysaccharides (EPSs)

EPSs are complex carbohydrate polymers secreted by probiotic bacteria such as *Lactobacillus*, *Bifidobacterium*, *Streptococcus*, *Weissella*, during metabolic processes [168]. Structurally, exopolysaccharides consist of repeating units of glucose, galactose, mannose and rhamnose, which can form homo or heteropolysaccharides, often linked by glycosidic bonds [169]. These polysaccharides can either be covalently anchored to the cell surface, forming capsular polysaccharides [170], or be secreted into the extracellular environment, leading to the formation of a mucilaginous layer. This latter form plays a critical role in the development of bacterial biofilms [170,171]. These types of polysaccharides can vary widely in composition, branching, and molecular weight, contributing to their diverse functional properties and activities [172].

One of the most notable is their bifidogenic activity, which refers to their ability to selectively stimulate the growth of specific members of the infant gut microbiota, specifically of *Bifidobacterium* species [173]. In infants, *Bifidobacterium* typically accounts for about 90% of intestinal bacteria [174].

The bifidogenic effects of EPSs arise from their complex carbohydrate structures, which are resistant to digestion in the upper gastrointestinal tract. Upon reaching the colon, these EPSs serve as prebiotics. Lv et al. (2024) isolated and purified EPSs from *Bifidobacterium animalis* subsp. Lactis SF (SF-EPS) from the feces sample of a healthy infant), and their probiotic potential was evaluated in vitro. SF-EPS regulated the gut microbiota by increasing the relative abundances of *Faecalibacterium*, *Anaerostipes*, and *Bifidobacterium*, while reducing the abundance of *Enterobacter* and *Klebsiella*. Furthermore, SF-EPS enhanced the production of SCFAs by intestinal microorganisms. These findings suggest that SF-EPS may serve as a potential prebiotic for use in functional foods [14].

Exopolysaccharides also exhibit potent biological activities, including antioxidant properties, free radical scavenging, and the reduction of oxidative stress [14,175]. Tarique et al. (2024) showed the antioxidant potential of EPSs from *Enterococcus faecium* and *S. thermophilus*. This potential could be due to the different sugars and their arrangements which can affect the ability of EPSs to interact with and neutralize free radicals [26].

EPSs also have immunoregulatory properties. EPSs from *Bifidobacterium longum* subsp. infantis E4 demonstrated significant immunomodulatory and anti-inflammatory effects in vitro. These EPSs enhanced macrophage activity and reduced inflammatory markers, indicating potential benefits for immune health in infants [176]. Additionally, some studies have found that EPSs from probiotics have other properties, such as antitumor activity [177,178,179], antibacterial activity [180], antiviral protection [181], and lipid regulation potential [182,183]. EPSs also enhance adherence and subsequent colonization of microflora on host cells [14].

In summary, their combination of different biological activities makes EPSs promising candidates for functional foods and therapeutic agents aimed at improving children’s health.

### 5.6. Neurotransmitters

#### 5.6.1. Gamma-Aminobutyric Acid (GABA)

GABA is a crucial neurotransmitter for children, playing a key role in brain development, emotional regulation, and cognitive functions like learning and memory. It helps maintain a balance between excitatory and inhibitory signals in the brain, which is important for reducing anxiety, promoting restful sleep, and managing stress [184,185,186]. Adequate GABA levels support healthy neural circuit formation, contribute to emotional stability, and help children with behavioral control, making GABA essential for overall mental and physical well-being [187,188]. Interestingly, these levels have been found to increase with age [189], and this difference in GABA levels seems to be related to how fast children can learn. Experimental results indicate children show more flexible GABA-related inhibitory processing compared to adults, allowing for quicker adaptation to stabilize learning [186].

While *Lactobacillaceae* species are recognized as primary producers of GABA, Bifidobacterium species have been identified as the most efficient GABA producers [187]. Notably, *gad* genes responsible for GABA synthesis are also found in other probiotic strains [190,191]. The production of GABA by these microorganisms has suggested a link between the gut microbiota and neurological health [191]. GABA is produced by the enzyme glutamate decarboxylase (GAD), which requires pyridoxal-5′-phosphate (PLP) and works through the irreversible α-decarboxylation of l-glutamate, consuming one cytoplasmic proton [192].

Bifidobacterium species such as *Bifidobacterium adolescentis*, *Bifidobacterium dentium*, and *Bifidobacterium longum* have demonstrated the capacity to synthesize GABA through the decarboxylation of glutamate by GAD enzymes. This process not only helps in maintaining the gut’s acid–base balance but also contributes to the pool of bioactive GABA within the host [190,193,194,195,196].

GABA-producing probiotic strains significantly impact the gut–brain axis, a communication network between the gastrointestinal tract and the central nervous system, involving the gut microbiota, immune system, enteric nervous system (ENS), and central nervous system [197]. GABA can modulate ENS activity—affecting gut motility, secretion, and blood flow [198]—and may influence the central nervous system via the vagus nerve, impacting neuropsychiatric conditions [199,200]. Increased GABA levels in the gut have been linked to reduced stress and anxiety-like behaviors [185], with specific bacteria like Bacteroides and GABA-producing Bifidobacterium strains playing roles in mental health and reducing systemic inflammation associated with mood disorders and neurodegenerative diseases [191,200].

Previous studies have demonstrated that the gut microbiota diversity in children with autism spectrum disorder (ASD) undergoes significant changes, with alterations in *Bifidobacterium* being linked to the severity of ASD [201]. Infants at higher risk for ASD have a decreased abundance of *Bifidobacterium* and an increased abundance of *Clostridium* and *Klebsiella* compared to those at lower risk. Additionally, fecal GABA levels were lower in infants with a higher likelihood of ASD, with GABA levels showing a positive correlation with Bifidobacterium [202].

Analyses on children with attention deficit hyperactivity disorder (ADHD) have found lower levels of GABA and reduced presence of lactic acid bacteria. These bacteria are known to be involved in the production of GABA, suggesting that the reduction of lactic acid bacteria in the gut of infants with ADHD could be associated with lower GABA levels [203,204].

Furthermore, clinical trials have suggested that probiotics containing GABA-producing *Bifidobacterium* or *Lactobacillaceae* species can be effective in treating gastrointestinal issues and enhancing overall mental health in children [205,206,207].

#### 5.6.2. Other Neurotransmitters Produced by Probiotics

In addition to GABA, probiotic microorganisms are gaining recognition for their capacity to produce various neurotransmitters, which can significantly influence host health via the gut–brain axis. A well-documented example is serotonin (5-HT), a neurotransmitter crucial for regulating mood, appetite, and sleep. Certain probiotic strains, such as *Enterococcus* and *Streptococcus*, have been shown to synthesize serotonin, suggesting their potential impact on emotional and psychological well-being [208].

Dopamine, another essential neurotransmitter associated with reward and motivation, is produced by probiotic strains like *Bacillus* and *Lactobacillus* species [209]. The production of dopamine by these probiotics could affect neurological functions and behaviors, highlighting a possible route through which to influence the host’s nervous system. Similarly, *Lactobacillus plantarum* is known to produce acetylcholine, which plays a critical role in learning, memory, and muscle activation [210]. The presence of acetylcholine in the gut underscores the complex communication pathways of the gut–brain axis.

Probiotics like *Escherichia coli* and *Bacillus subtilis* have also demonstrated the ability to synthesize norepinephrine, a neurotransmitter involved in alertness and the body’s “fight or flight” response [211]. This ability suggests that these microorganisms could help modulate stress and mood responses. Additionally, *Lactobacillus reuteri* is known to produce histamine, which is critical for immune responses and gut motility [212]. The production of glutamate, an excitatory neurotransmitter important for synaptic plasticity, has been observed in strains of *Lactobacillus* and *Bifidobacterium* [213].

Overall, the capacity of probiotics to synthesize these neurotransmitters supports their potential role in managing neuropsychiatric and gastrointestinal disorders by modulating the gut–brain axis, providing a promising avenue for therapeutic intervention.

### 5.7. Bioactive Postbiotic Fractions

Postbiotics, defined by the International Scientific Association for Probiotics and Prebiotics (ISAPP) as “inanimate microorganisms and/or their components that confer health benefits to the host”, include microbial cells, metabolites, and fermentation byproducts [214]. Postbiotics are key mediators of microbiota–host interactions. Unlike probiotics, which are live microorganisms, postbiotics do not contain live bacteria but consist of beneficial byproducts released during the microorganisms’ life cycle [215,216]. Postbiotics can be classified based on their composition, including lipids (e.g., butyrate, propionate), proteins (e.g., lactocepin), carbohydrates (e.g., polysaccharides, teichoic acids), vitamins, organic acids (e.g., lactic acid), and complex molecules (e.g., lipoteichoic acids) [215,216,217]. Additionally, they can be categorized by their physiological functions, such as immunomodulatory, anti-inflammatory, hypocholesterolemic, anti-obesity, antihypertensive, anti-proliferative, and antioxidant effects (Figure 4) [218,219]. These compounds are widely present in foods such as yogurt, kefir, and pickled vegetables, and can also be intentionally applied in functional foods such as infant formulas [215,216,220].

The significance of postbiotics lies in their potential health benefits and greater stability compared to probiotics. Because postbiotics lack live bacteria, they are more stable and have a longer shelf life, making them easier to store and use in therapeutic and nutritional applications [221]. The unique structure of postbiotics can exert a range of beneficial effects on the host through diverse cellular and molecular mechanisms. Postbiotics play a crucial role in children’s health by supporting gut balance, boosting immunity, and promoting digestive health. They help reduce inflammation, enhance nutrient absorption, and protect against infections, contributing to overall well-being in growing children [215,222]. Additionally, the use of postbiotics is associated with fewer safety concerns than probiotics, as there is no risk of infection from live bacteria. This makes them particularly suitable for vulnerable populations, including infants, the elderly, and immunocompromised individuals [215,216,223].

Postbiotics exhibit significant immunomodulatory and anti-inflammatory effects, playing a critical role in enhancing both innate and adaptive immune responses. One of the key mechanisms by which postbiotics exert immunomodulatory effects is through the activation of immune receptors such as Toll-like receptors (TLRs). For instance, heat-inactivated *Lactobacillus casei* has been shown to enhance macrophage-mediated innate immunity by increasing the transcription of TLRs (TLR2, TLR3, TLR4, and TLR9) and stimulating pro-inflammatory cytokines, which strengthen the body’s defense against infections [224]. Moreover, postbiotics derived from *Lactobacillus gasseri* TMC0356 have demonstrated a more potent effect on immune activity than their probiotic counterparts, inducing higher levels of interleukin-12 (IL-12) in macrophages [225].

The anti-inflammatory properties of postbiotics are equally noteworthy. By regulating cytokine production, postbiotics can help reduce inflammatory responses. For example, supernatants of *Faecalibacterium prausnitzii* have been shown to alleviate colitis in mice by increasing the production of the anti-inflammatory cytokine IL-10 while reducing the pro-inflammatory cytokine IL-12, likely through the inhibition of NF-κB activation [226]. Additionally, *Lactobacillus paracasei* B21060 postbiotics have demonstrated protective effects against inflammation caused by *Salmonella* in human colon tissues [227]. Another example of the immunomodulatory action of postbiotics involves *Lactobacillus reuteri* 17938, which promotes the production of the anti-inflammatory cytokine IL-10 in dendritic cells, leading to an enhanced regulatory T-cell response [228].

Recent research has increasingly focused on the role of fermented infant formulas and their postbiotic components in enhancing infant health, particularly in reducing the severity of gastrointestinal and allergic conditions. A study by Béghin et al. (2021) explored the effects of a fermented infant formula (FF) with *Bifidobacterium breve* C50 and *Streptococcus thermophilus* O65 combined with prebiotic oligosaccharides on gut microbiota composition and immune function in healthy term infants. The findings indicated that this combination led to a gut microbiota composition and metabolic activity more similar to that of breastfed infants, with a significant increase in secretory IgA (SIgA) levels, highlighting its potential to enhance early immune defense [229]. In contrast, a study by Thibault et al. (2004) focused on the impact of an FF with the same probiotic bacteria on the incidence and severity of acute diarrhea in healthy infants aged 4 to 6 months [230].

In summary, postbiotics offer a range of immunomodulatory and anti-inflammatory benefits, making them promising therapeutic agents. Their ability to regulate immune responses, reduce inflammation, and support overall immune health highlights their potential in managing diverse health [215,231,232].

## 6. Clinical Applications and Health Implications

A comprehensive analysis published by Dronkers et al. in 2020 on 1341 studies retrieved from the ClinicalTrials.gov database (using the search term “probiotics”) showed that 56% were conducted in the USA or Europe; around 100 studies have been registered annually since 2010, and this number has been increasing in recent years [233]. The vast majority of these studies were interventional (95.6%), but almost half were in healthy participants (43.8%), and only 31.8% of those studies were in children (from birth to 17 years old). Of the 852 studies that could be analyzed, *Lactobacillus rhamnosus* GG (LGG) was the probiotic strain most frequently registered (146 studies), followed by *Bifidobacterium animalis* ssp. *lactis* BB12 with 55 studies, while VSL#3, a consortium of three different *Bifidobacteria*, four *Lactobacillus*, and one *Streptococcus thermophilus* strains, was the most registered multispecies preparation, featuring in 74 studies.

Over the last five years, ClinicalTrials.gov has listed 730 studies, of which 167 were in children and 157 were interventional studies in children. Two studies were withdrawn (one in colic and other in peanut allergy); five were prematurely terminated (two in respiratory infections, one in atopic dermatitis, one in surgical procedures and one in healthy children), and forty-eight were completed, addressing a range of conditions including allergic diseases, respiratory infections, cystic fibrosis, obesity, ADHD, gastrointestinal diseases, and autism spectrum disorder (ASD). This indicates that 84 studies (53.5%) are registered but still ongoing. Despite this focus, the volume of published clinical trials in children has been decreasing according to the Pubmed.gov database, from 69 publications in 2020 and 2021 to 55 articles in 2022 and just 46 in the last year.

In recent years, the role of probiotics in managing ASD has been studied, as the potential benefits of probiotics have been suggested by explorations of the gut–brain axis, thus offering hope for improved management strategies [234]. However, a meta-analysis published in the Journal of Medical Microbiology in 2022 showed no significant benefit of probiotics for ASD treatment [235]. This study highlighted that the lack of standardization in trials—such as variations in strains, dosages, and protocols—complicates our ability to conduct comprehensive meta-analyses with robust findings.

### Current Practice Guidelines

Recommendations for the clinical application of probiotics still show discrepancies between existing guidelines, particularly those for acute gastroenteritis in children, as summarized in Table 2.

American Academy of Pediatrics (AAP): Probiotics such as *L. rhamnosus* GG and *S. boulardii* are recommended for reducing the duration of acute gastroenteritis in children. However, routine use in healthy children is not broadly advised. Probiotics may help prevent antibiotic-associated diarrhea and necrotizing enterocolitis (NEC) in preterm infants, though caution is advised in immunocompromised patients [236].

European Society for Pediatric Gastroenterology, Hepatology and Nutrition (ESPGHAN): Probiotics for conditions like acute gastroenteritis and prevention of NEC in preterm infants are supported, but strain-specific efficacy and proper quality control are emphasized [237].

The British Society of Pediatric Gastroenterology, Hepatology and Nutrition (BSPGHAN) aligns with ESPGHAN, emphasizing selective use of probiotics for gastrointestinal issues and the importance of strain-specific evidence in children [236].

For instance, the American Gastroenterological Association (AGA) is against the use of probiotics [236,238]. These inconsistencies arise from several factors, including variability in literature review processes—such as incomplete searches or differing inclusion criteria—which can lead to conflicting recommendations. Additionally, some guidelines may place more weight on single randomized controlled trials (RCTs), while others incorporate a broader evidence base [239].

Differences in study populations, including socioeconomic levels, medical issues, and geographical features, also influence these recommendations. Potential conflicts of interest (COI) bias from industry sponsorship may further impact guideline development. Lastly, the evolving nature of research means that guidelines may not always reflect the most current evidence. These factors highlight the need for more standardized approaches and regular updates to clinical guidelines to support better decision making and foster ongoing research [239].

**Table 2 nutrients-16-03706-t002:** Current Guidelines on Probiotics for most common conditions (modified from [240]).

Disorder	Probiotic Strain	Recommended Dose	Evidence Level **
Acute gastroenteritis [reduced the risk of diarrhea lasting ≥48 h; reduced the mean duration of diarrhea [241]]	Probiotics as a general group	N/A	1
	*L. rhamnosus* GG [242]	≥10^10^ cfu/day, for 5–7 days	1
	*S. boulardii* * [243]	250–750 mg/day, for 5–7 days	1
	*L. reuteri* DSM 17938 [244]	1 × 10^8^ to 4 × 10^8^ cfu/day, for 5 days	1
	*L. rhamnosus* 19070-2 & *L. reuteri* DSM 12246 [245]	2 × 10^10^ cfu for each strain/day, for 5 days	1
	*B. lactis* B94 [246]	5 × 10^10^ cfu once daily, for 5 days	3
	*L. paracasei* B21060 [247]	2.5 × 10^9^ cfu, twice daily, for 5 days	3
	*L. rhamnosus* strains 573L/1; 573L/2; 573L/3 [248]	1.2 × 10^10^ cfu, twice daily, for 5 days	3
	*L. delbrueckii* var. *bulgaricus*, *L. acidophilus*, *S. thermophilus*, *B. bifidum* (LMG-P17550, LMG-P 17549, LMG-P 17503, LMG-P 17500) [249]	10^9^ cfu, 10^9^ cfu, 10^9^ cfu, 5 × 10^8^ cfu/dose, for 5 days	3
	*B. lactis* Bi-07, *L. rhamnosus* HN001, and *L. acidophilus* NCFM [250]	Then, 1 × 10^10^ cfu once a day, for the duration of diarrhea plus 7 days	3
Prevention of AAD (reduced risk of AAD [251])	Probiotics as a general group	N/A	1
	*S. boulardii* * [252]	≥5 billion cfu per day, for the duration of antibiotic treatment	1
	*L. rhamnosus GG* [253]	≥5 billion cfu per day, for the duration of antibiotic treatment	1
	Multispecies probiotic (*Bifidobacterium bifidum W23*, *B. lactis W51*, *Lactobacillus acidophilus W37*, *Lactobacillus acidophilus W55*, *Lacticaseibacillus paracasei W20*, *Lactoplantibacillus plantarum W62*, *Lacticaseibacillus rhamnosus W71*, and *Ligilactobacillus salivarius W24*) [254]	10 billion cfu per day, for the duration of antibiotic treatment and for 7 days after	3
	*L. rhamnosus* (strains E/N, Oxy, and Pen) [255]	2 × 10^10^ cfu, twice daily, for the duration of antibiotic treatment	3
Prevention of *C. difficile* diarrhea	*S. boulardii* * [252]	250–500 mg	1
Prevention of nosocomial diarrhea	*L. rhamnosus GG* [256,257]	At least 10^9^ cfu/day, for the duration of the hospital stay	1
Prevention of necrotizing enterocolitis [258,259,260]	Systematic reviews and meta-analyses (>10,000 neonates) of RCTs		1
	*L. rhamnosus GG* [261]	From 1 × 10^9^ to 6 × 10^9^ cfu	1
	*B. infantis* BB-02, *B. lactis* BB-12, and *S. thermophilus* TH-4 [261]	3.0 to 3.5 × 10^8^ cfu (of each strain)	1
	*B. animalis* subsp. *lactis* Bb-12 or B94 [261]	5 × 10^9^ cfu	3
	*L. reuteri* ATCC 55730 * or DSM 17938 * this strain is no longer available. [261,262]	1 × 10^8^ cfu (various regimens)	1
	*B. longum* subsp. *infantis* ATCC 15697 *+ L. acidophilus* ATCC 4356 [262,263]	125 mg/kg/dose twice daily with breast milk until discharge	3
	*B. longum* subsp. *longum* 35624 + *L. rhamnosus* GG [263]	5 × 10^8^ cfu and 5 × 10^8^ cfu, respectively	3
*Helicobacter pylori* infection [264,265,266,267,268]	Probiotics as a general group		1
	*S. boulardii ** [269,270]	500 mg	1
Infantile colic [271,272,273,274,275,276]	Probiotics as a general group	N/A	1
	*L. reuteri* DSM 17938 [277,278]	10^8^ cfu/day for at least 21 days	1
	*B. lactis* Bb12 [279,280]	1 × 10^9^ cfu/day, for 21–28 days	2
	*L. rhamnosus* 19070-2 and *L. reuteri* 12246 [281]	250 × 10⁶ cfu, respectively, for 28 days	3
	*L. paracasei* DSM 24733, *L. plantarum* DSM 24730, *L. acidophilus* DSM 24735, *L. delbrueckii* subsp. *bulgaricus* DSM 24734), *B. longum* DSM 24736, *B. breve* DSM 24732, and *B. infantis* DSM 24737, and *S. thermophilus* DSM 24731 [282]	5 billion cfu, for 21 days	3
Infantile colic prevention	*L. reuteri* DSM 17938 [283]	10^8^ cfu/day, to newborns each day for 90 days	1
Functional abdominal pain/IBS	*L. reuteri* DSM 17938 [284,285]	10^8^ cfu to 2 × 10^8^ cfu/day	1
	*L. rhamnosus GG* [284,286]	10^9^ cfu to 3 × 10^9^ cfu twice daily	1
Ulcerative colitis [287]	Probiotics as a group	N/A	1
	A mixture of 8 strains (*L. paracasei* DSM 24733, *L. plantarum* DSM 24730, *L. acidophilus* DSM 24735, *L. delbrueckii* subsp. *bulgaricus* DSM 24734, *B. longum* DSM 24736, *B. infantis* DSM 24737, *B. breve* DSM 24732, and *S. thermophilus* DSM 247), as adjuvant therapy or in those intolerant to 5-ASA [288]	Daily dosages: 4–6 y (17–23 kg): 450 billion; 7–9 y (24–33 kg): 900 billion; 11–14 y (34–53 kg): 1350 billion; 15–17 y (54–66 kg): 1800 billion.	3
Pouchitis	A mixture of 8 strains (*L. paracasei* DSM 24733, *L. plantarum* DSM 24730, *L. acidophilus* DSM 24735, *L. delbrueckii* subsp. *bulgaricus* DSM 24734, *B. longum* DSM 24736, *B. infantis* DSM 24737, *B. breve* DSM 24732, and *S. thermophilus* DSM 247) [289,290]	Daily dosages: 4–6 y (17–23 kg): 450 billion; 7–9 y (24–33 kg): 900 billion; 11–14 y (34–53 kg): 1350 billion; 15–17 y (54–66 kg): 1800 billion.	3

* Most studies with the strain *S. boulardii* CNCM I-745; ** (1) systematic review or meta-analysis of randomized control trials; (2) randomized control trials; (3) quasi-experimental studies; (4) non-experimental studies.

## 7. Challenges of Using Bioactive Molecules from Probiotics for Pediatric Diseases

The use of probiotics in children, particularly in the most vulnerable—such as preterm infants, immunocompromised, or those with underlying health conditions—is still limited by safety concerns. Although many studies confirm the safety of probiotics, there have been isolated cases of sepsis or gastrointestinal mucormycosis linked to contaminated probiotic products, highlighting the need for stringent quality control measures during manufacturing [291]. Furthermore, the reported risk of cross-colonization with probiotic strains in neonatal intensive care units (NICUs) suggested that despite probiotics offered potential benefits, they must be administered cautiously, supported by comprehensive safety assessments [291]. The Agency for Healthcare Research and Quality reviewed 622 studies on probiotic safety and found that while there is no evidence of increased risk from randomized controlled trials, the literature lacks systematic reporting on adverse events, making it difficult to assess rare risks with confidence [292]. Therefore, it is crucial to implement standardized protocols for safety outcomes in clinical trials, ensuring comprehensive and transparent data collection. Enhanced guidelines would enable a more accurate assessment of the benefits/risks associated with probiotic use and support informed decision making in both research and clinical practice.

Translating promising findings from preclinical and clinical studies into clinical guidelines is an imminent challenge. Research conducted in diverse geographic regions may not be generalizable due to variations in genetics, diet, sanitation, and endemic enteropathogens [293]. Systematic reviews have thus far failed to recommend specific strains due to inconsistency in results and lack of standardized protocols regarding species, dosage, and administration duration [233]. Variations in study designs, including differences in probiotic strains, dosages, and treatment durations, have led to inconsistent results and have hindered the development of clear guidelines. Moreover, the absence of standardized protocols and regulatory frameworks complicates the approval and adoption of probiotics for pediatric use. To overcome these obstacles, more well-designed randomized controlled trials (RCTs) and meta-analyses are needed to establish evidence-based guidelines for probiotic use in children.

Clinical trials in pediatric populations present additional ethical challenges and special regulatory compliance. Moreover, differences in diet, genetics, and environmental factors across different populations limit the generalization of the research findings. Regulatory agencies should develop frameworks that promote innovation while ensuring safety and efficacy, facilitating global standardization in probiotic research and applications.

A significant limitation in many studies is the underexplored interaction between different probiotic species, which may work synergistically. Most preclinical and clinical investigations focus on isolated strains, neglecting the complex interactions within the entire gut microbiota. The gut microbiome is a unique ecosystem for each individual, harboring diverse microorganisms. Interactions among multiple probiotic species can enhance therapeutic efficacy through synergistic mechanisms [294,295].

Current research primarily emphasizes individual probiotics, yet the potential for multi-strain synergism warrants further investigation. Combining strains with complementary roles, such as one that strengthens gut barrier integrity and another that modulates immune responses, could improve treatment strategies for complex pediatric conditions like inflammatory bowel disease (IBD) and allergies [296,297,298]. However, studying these interactions requires long-term research to navigate the microbiome’s complexities.

Furthermore, probiotics do not function in isolation; their introduction can alter the native microbial community, yielding both positive and negative effects. Understanding these dynamics is crucial for developing effective, personalized probiotic therapies. Regulatory agencies should promote frameworks that foster innovation while ensuring safety and efficacy in probiotic research.

## 8. Future Perspectives and Opportunities

Successful integration of probiotic-derived bioactive molecules in the pediatric population is also an opportunity to enhance treatment options and potentially improve outcomes. In addressing the previously discussed challenges, we can make substantial strides toward safer, more effective, and widely accepted probiotic-based therapies for pediatric populations.

### 8.1. Fostering of Clinical Research

The current situation of clinical trials with probiotics in the pediatric population reveals a growing interest in the field, while the decrease in the number of publications might indicate stronger review criteria for the available data. Future allocation of resources, multidisciplinary teams, and active collaboration are essential to strengthen clinical research oriented to answer relevant and important questions.

### 8.2. Evidence-Based Clinical Guidelines

The current data available and their variability offer an opportunity to foster broader and global collaborations among healthcare providers, their institutions, research-oriented organizations, academia, and industry. Therefore, creating stronger evidence for the use of probiotics in children should be a must. Development of updated guidelines (based on evidence) that consider regional differences in diet, genetics, and environmental factors will further facilitate personalized treatments.

### 8.3. Strengthening Ethical Frameworks for Pediatric Research

Pediatric research presents unique opportunities, despite its ethical challenges, which can be addressed by developing innovative frameworks that balance patient protection with medical progress. Regulatory agencies can set international ethical standards for probiotic trials, including informed consent and independent ethical reviews. Transparent data sharing, and international collaboration can build trust and accelerate the transition from research to clinical practice.

### 8.4. Enhancing Professional Education and Interdisciplinary Collaboration

Promoting literacy in basic sciences (e.g., biochemistry and microbiology) among healthcare professionals through interdisciplinary workshops, seminars, and updated curricula fosters innovation and informed decision making. Integrating probiotic science into medical education and offering continuous professional development may lead to the discovery of new probiotic strains or bioactive compounds with unique benefits for children, thus enhancing clinical practice and research.

## Figures and Tables

**Figure 1 nutrients-16-03706-f001:**
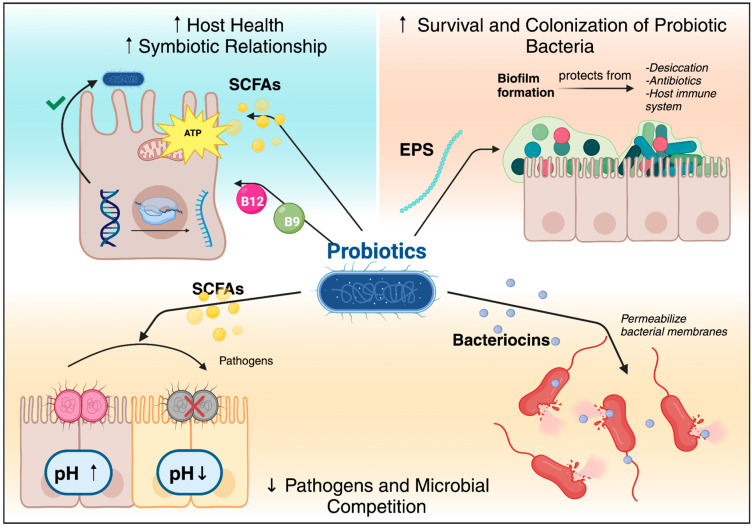
Role of metabolites in the adaptation of probiotic bacteria within the gut environment. This figure depicts how probiotic metabolites benefit both the bacteria and the host: 1. symbiotic interaction; 2. self-protection; and 3. pathogen inhibition. EPSs: exopolysaccharides; SCFAs: short-chain fatty acids; pH: potential of hydrogen; ATP: adenosine triphosphate.

**Figure 2 nutrients-16-03706-f002:**
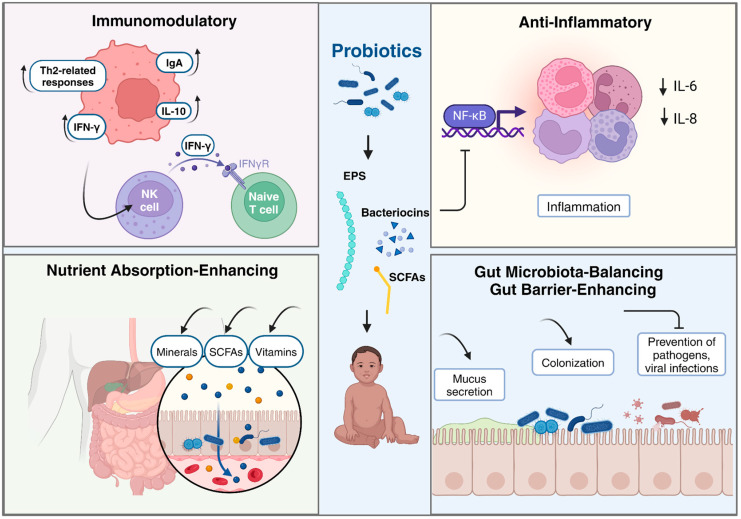
Role of probiotic metabolites in children’s health. Short-chain fatty acids (SCFAs), exopolysaccharides (EPSs), Immunoglobulin A (IgA), interleukin-10 (IL-10), and interferon-gamma (IFN-γ) in immune cells. Nuclear factor kappa B (NF-κB) activation leading to reduced levels of interleukin-6 (IL-6) and interleukin-8 (IL-8).

**Figure 3 nutrients-16-03706-f003:**
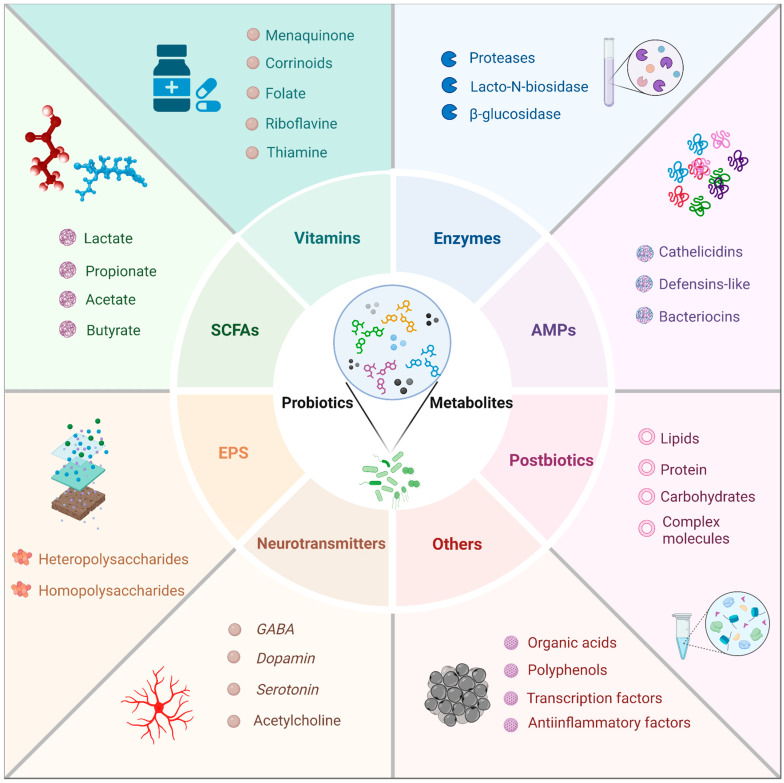
Major metabolites produced by probiotic bacteria in children. SCFAs: short-chain fatty acids; EPSs: exopolysaccharides; GABA: gamma-aminobutyric acid; AMPS: antimicrobial peptides.

**Figure 4 nutrients-16-03706-f004:**
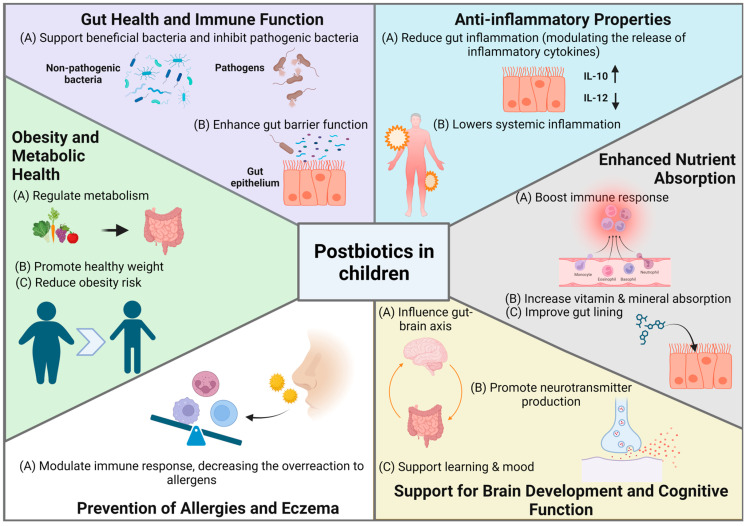
Postbiotics and their potential positive effects in children.

**Table 1 nutrients-16-03706-t001:** Key vitamins produced by probiotic bacteria.

Probiotic Strain	Vitamin Produced	Mechanism
*Lactobacillus fermentum* [122]	Folate (B9), Vitamin B12 (Cobalamin)	Synthesis of folate and B12
*Lactobacillus reuteri* (DCM 20016, JCM1112, CRL1324, CRL1327) [31]	Corrinoids (related to Vitamin B12)	Production of corrinoids
*Lactobacillus acidophilus* (ATCC314, FTDC 8833) [121]	Riboflavin (B2)	Enhances riboflavin production
*Streptococcus thermophilus* (ABM5097) [15]	5-Methyltetrahydrofolate (5-MTHF) (Folate)	Increases folate production
*Lactobacillus rhamnosus GG* [31]	Folate (B9), Riboflavin (B2), Thiamine (B1)	Produces and releases folate and riboflavin efficiently; low production of intracellular thiamine
*Bacillus clausii* [131]	Vitamin K2 (Menaquinone)	Production of vitamin K2
*Lactococcus lactis* [132]	Vitamin K2 (Menaquinone)	Production of vitamin K2
*Enterobacter agglomerans*, *Serratia marcescens*, *Enterococcus faecium* [133]	Menaquinones (Vitamin K2)	Contributes to vitamin K production in the neonatal gut
Various strains (*Lactobacillus*, *Bifidobacterium*) [123]	B Vitamins (B1, B2, B3, B5, B6, B7, B9, B12), Vitamin K	Utilizes oligosaccharides to enhance hydrophobicity, auto-aggregation, and biofilm formation, thus improving B vitamin production
*Lactobacillus gasseri* (FTDC 8131) [135]	Riboflavin (B2)	Interacts with riboflavin; context suggests strain-dependent variability in production or consumption of the vitamin
Bifidobacterium strains (*B. longum*, *B. bifidum*) [31]	Thiamine (B1)	Low but significant production of intracellular thiamine without extracellular synthesis; does not produce folates or riboflavin
Children consuming probiotics [98]	Vitamin D, Vitamin A	Probiotics enhance absorption and serum concentrations of vitamins

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
