# Peer review of "The Impact of Bioactive Molecules from Probiotics on Child Health: A Comprehensive Review"

_nutrients, 2024, doi:10.3390/nu16213706_

Round 1

Reviewer 1 Report

Comments and Suggestions for Authors

The comprehensive review is well-written and organized and covers all aspects of the impact of bioactive molecules from probiotics on child health, focusing on gut microbiota, immune function, and overall development. Key metabolites such as SCFAs, bacteriocins, EPS, vitamins, and GABA are highlighted for their ability to maintain gut health and support neurodevelopment. Specific probiotics and their metabolites have shown promise in reducing gastrointestinal disorders, enhancing immune responses, and decreasing the incidence of allergies and respiratory infections in children. Postbiotics from probiotic fermentation offer benefits such as improved gut barrier function and enhanced nutrient absorption with fewer safety concerns compared to live probiotics.

Thanks to the authors for their time and effort to compile this comprehensive review. I only recommend making the review more concise.

Author Response

Dear Reviewer,

Thank you for your positive feedback on our manuscript and for recognizing the comprehensive nature of our review on the impact of bioactive molecules from probiotics on child health. We appreciate your suggestion to make the review more concise.

In response, we have streamlined the manuscript by focusing on the most critical and impactful findings and condensing sections that may have been overly detailed. Specifically, Sections 3 and 5 have been merged, as both relate to health benefits in children, and the section on bioactive postbiotic fractions has been made more concise to eliminate redundancy both sections were highlighted for your revision. These changes ensure that the review remains thorough while being more concise and easier for readers to navigate.

We are grateful for your thoughtful recommendation and have incorporated these revisions accordingly.

Reviewer 2 Report

Comments and Suggestions for Authors

The present comprehensive review is not only interesting and helpful from a clinical perspective, but also significantly important. The authors' analysis of the efficacy of probiotics on child health, focusing on their roles in modulating gut microbiota, is a crucial contribution to the field.

The authors could add another limitation for all reported studies.

There is an exciting potential for interaction between the different species of probiotics, and they could act synergistically in many cases, a field that remains unexplored in depth and for a long duration. However, in many preclinical and clinical studies, one or more probiotics have been investigated, not the whole gut microbiota.

The authors rightly emphasize the need to consider regional differences in diet, genetics, and environmental  factors when developing updated guidelines for the use of probiotics in children. This personalized approach, based on evidence, is crucial for effective treatments.

The methodology is appropriate. The manuscript is well written, and the discussion/conclusions are acceptable.

Overall, data are of interest.

Comments on the Quality of English Language

none

Author Response

Thank you for your insightful and encouraging comments regarding our manuscript. We appreciate your acknowledgment of the clinical significance of our review and the analysis of probiotics' efficacy on child health.

In response to your suggestion, we have added a new limitation to Section 7, "Challenges of Using Bioactive Molecules from Probiotics for Pediatric Diseases." We have highlighted the exciting potential for synergistic interactions between different probiotic species, emphasizing that this area remains underexplored in many preclinical and clinical studies, which typically investigate only one or a few probiotics rather than the entire gut microbiota.

Thank you once again for your thoughtful review.